# Area Properties of Strictly Convex Curves

**Dong-Soo Kim [1], Young Ho Kim [2,\*] and Yoon-Tae Jung [3]**

[1]   Department of Mathematics, Chonnam National University, Gwangju 61186, Korea; dosokim@chonnam.ac.kr
[2]   Department of Mathematics, Kyungpook National University, Daegu 41566, Korea
[3]   Department of Mathematics, Chosun University, Gwangju 61452, Korea; ytajung@chosun.ac.kr
\*   Correspondence: yhkim@knu.ac.kr; Tel.: +82-53-950-5882

**Abstract:** We study functions defined in the plane $\mathbb{E}^2$ in which level curves are strictly convex, and investigate area properties of regions cut off by chords on the level curves. In this paper we give a partial answer to the question: Which function has level curves whose tangent lines cut off from a level curve segment of constant area? In the results, we give some characterization theorems regarding conic sections.

**Keywords:** Archimedes; level curve; chord; conic section; strictly convex plane curve; curvature; equiaffine transformation

## 1. Introduction

The most well-known plane curves are straight lines and circles, which are characterized as the plane curves with constant Frenet curvature. The next most familiar plane curves might be the conic sections: ellipses, hyperbolas and parabolas. They are characterized as plane curves with constant affine curvature ([1], p. 4).

The conic sections have an interesting area property. For example, consider the following two ellipses given by $X_k = g^{-1}(k)$ and $X_l = g^{-1}(l)$ with $l > k > 0$, where

$$g(x,y) = \frac{x^2}{a^2} + \frac{y^2}{b^2}, a, b > 0.$$

For a fixed point $p$ on $X_k$, we denote by $A$ and $B$ the points where the tangent to $X_k$ at $p$ meets $X_l$. Then the region $D$ bounded by the ellipse $X_l$ and the chord $AB$ outside $X_k$ has constant area independent of the point $p \in X_k$.

In order to give a proof, consider a transformation $T$ of the plane $\mathbb{E}^2$ defined by

$$T = \begin{pmatrix} b/\sqrt{ab} & 0 \\ 0 & a/\sqrt{ab} \end{pmatrix}.$$

Then $X_k$ and $X_l$ are transformed to concentric circles of radius $\sqrt{abk}$ and $\sqrt{abl}$, respectively; the tangent at $p$ to the tangent at the corresponding point $p'$. Since the transformation $T$ is equiaffine (that is, area preserving), a well-known property of concentric circles completes the proof.

For parabolas and hyperbolas given by $g(x,y) = y^2 - 4ax, a \neq 0$ and $g(x,y) = x^2/a^2 - y^2/b^2, a, b > 0$, respectively, it is straightforward to show that they also satisfy the above mentioned area properties. For a proof using 1-parameter group of equiaffine transformations, see [1], pp. 6–7.

Conversely, it is reasonable to ask the following question.

**Question.** Are there any other level curves of a function $g : \mathbb{R}^2 \to \mathbb{R}$ satisfying the above mentioned area property?

A plane curve $X$ in the plane $\mathbb{E}^2$ is called 'convex' if it bounds a convex domain in the plane $\mathbb{E}^2$ [2]. A convex curve in the plane $\mathbb{E}^2$ is called 'strictly convex' if the curve has positive Frenet curvature $\kappa$ with respect to the unit normal $N$ pointing to the convex side. We also say that a convex function $f : \mathbb{R} \to \mathbb{R}$ is 'strictly convex' if the graph of $f$ is strictly convex.

Consider a smooth function $g : \mathbb{R}^2 \to \mathbb{R}$. We let $R_g$ denote the set of all regular values of the function $g$. We suppose that there exists an interval $S_g \subset R_g$ such that for every $k \in S_g$, the level curve $X_k = g^{-1}(k)$ is a smooth strictly convex curve in the plane $\mathbb{E}^2$. We let $S_g$ denote the maximal interval in $R_g$ with the above property. If $k \in S_g$, then there exists a maximal interval $I_k \subset S_g$ such that each $X_{k+h}$ with $k + h \in I_k$ lies in the convex side of $X_k$. The maximal interval $I_k$ is of the form $(k, a)$ or $(b, k)$ according to whether the gradient vector $\nabla g$ points to the convex side of $X_k$ or not.

As examples, consider the two functions $g_i : \mathbb{R}^2 \to \mathbb{R}, i = 1, 2$ defined by $g_i(x, y) = y^2 + \epsilon_i a^2 x^2$ with positive constant $a$, $\epsilon_i = (-1)^i$. Then, for the function $g_1$ we have $R_{g_1} = \mathbb{R} - \{0\}$, $S_{g_1} = (0, \infty)$ or $(-\infty, 0)$, $I_k = (k, \infty)$ if $k > 0$, and $I_k = (-\infty, k)$ if $k < 0$. For $g_2$, we get $R_{g_2} = S_{g_2} = (0, \infty)$ and $I_k = (0, k)$ with $k \in S_{g_2}$.

For a fixed point $p \in X_k$ with $k \in S_g$ and a small $h$ with $k + h \in I_k$, we consider the tangent line $t$ to $X_k$ at $p \in X_k$ and the closest tangent line $\ell$ to $X_{k+h}$ at a point $v \in X_{k+h}$, which is parallel to the tangent line $t$. We let $\mathcal{A}_p^*(k, h)$ denote the area of the region bounded by $X_k$ and the line $\ell$ (See Figure 1).

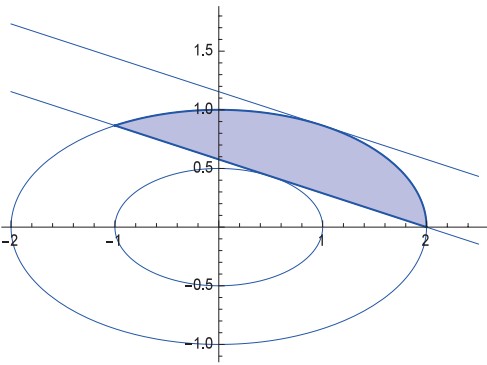

**Figure 1.** $\mathcal{A}_p^*(1, -3/4)$ for $p = (1, \sqrt{3}/2), v = (1/2, \sqrt{3}/4)$ and $g(x, y) = x^2/4 + y^2$.

In [3], the following characterization theorem for parabolas was established.

**Proposition 1.** *We consider a strictly convex function $f : \mathbb{R} \to \mathbb{R}$ and the function $g : \mathbb{R}^2 \to \mathbb{R}$ given by $g(x, y) = y - f(x)$. Then, the following conditions are equivalent.*

1.  *For a fixed $k \in \mathbb{R}$, $\mathcal{A}_p^*(k, h)$ is a function $\phi_k(h)$ of only $h$.*

2.  *Up to translations, the function $f(x)$ is a quadratic polynomial given by $f(x) = ax^2$ with $a > 0$, and hence every level curve $X_k$ of $g$ is a parabola.*

In the above proposition, we have $R_g = S_g = \mathbb{R}$ and $I_k = (k, \infty)$.

In particular, Archimedes proved that every level curve $X_k$ (parabola) of the function $g(x, y) = y - ax^2$ in the Euclidean plane $\mathbb{E}^2$ satisfies $\mathcal{A}_p^*(k, h) = ch\sqrt{h}$ for some constant $c$ which depends only on the parabola [4].

In this paper, we investigate the family of strictly convex level curves $X_k, k \in S_g$ of a function $g : \mathbb{R}^2 \to \mathbb{R}$ which satisfies the following condition.

$(\mathcal{A}^*)$: For $k \in S_g$ with $k + h \in I_k$, $\mathcal{A}_p^*(k, h)$ with $p \in X_k$ is a function $\phi_k(h)$ of only $k$ and $h$.

In order to investigate the family of strictly convex level curves $X_k, k \in S_g$ of a function $g : \mathbb{R}^2 \to \mathbb{R}$ satisfying condition $(\mathcal{A}^*)$, first of all, in Section 2 we introduce a useful lemma which reveals a relation between the curvature of level curves and the gradient of the function $g$ (Lemma 3 in Section 2).

Next, using Lemma 3, in Section 3 we establish the following characterizations for conic sections.

**Theorem 1.** *Let $f : \mathbb{R} \to \mathbb{R}$ be a smooth function. We let $g$ denote the function defined by $g(x, y) = y^a - f(x)$, where $a$ is a nonzero real number with $a \neq 1$. Suppose that the level curves $X_k(k \in S_g)$ of $g$ in the plane $\mathbb{E}^2$ are strictly convex. Then the following conditions are equivalent.*

1. *The function $g$ satisfies $(\mathcal{A}^*)$.*
2. *For $k \in S_g$, $\kappa(p)|\nabla g(p)|^3 = c(k)$ is constant on $X_k$, where $\kappa(p)$ denotes the curvature of $X_k$ at $p \in X_k$.*
3. *We have $a = 2$ and the function $f$ is a quadratic function. Hence, each $X_k$ is a conic section.*

In case the function $f$ ($-f$, resp.) is itself a non-negative strictly convex function, Theorem 1 is a special case ($n = 1$) of Theorem 2 (Theorem 3, resp.) in [5].

In Section 4 we prove the following.

**Theorem 2.** *Let $f : \mathbb{R} \to \mathbb{R}$ be a smooth function. For a rational function $j(y)$ in $y$, we let $g$ denote the function defined by $g(x, y) = f(x) + j(y)$. Suppose that the level curves $X_k(k \in S_g)$ of $g$ in the plane $\mathbb{E}^2$ are strictly convex. Then the following conditions are equivalent.*

1. *The function $g$ satisfies $(\mathcal{A}^*)$.*
2. *For $k \in S_g$, $\kappa(p)|\nabla g(p)|^3 = c(k)$ is constant on $X_k$, where $\kappa(p)$ denotes the curvature of $X_k$ at $p \in X_k$.*
3. *Both of the functions $j(y)$ and $f(x)$ are quadratic. Hence, each $X_k$ is a conic section.*

When the function $g$ is homogeneous, in Section 5 we prove the following characterization theorem for conic sections.

**Theorem 3.** *Let $g : \mathbb{R}^2 \to \mathbb{R}$ be a smooth homogeneous function of degree $d$. Suppose that the level curves $X_k$ of $g$ with $k \in S_g$ in the plane $\mathbb{E}^2$ are strictly convex. Then the following conditions are equivalent.*

1. *The function $g$ satisfies $(\mathcal{A}^*)$.*
2. *For $k \in S_g$, $\kappa(p)|\nabla g(p)|^3 = c(k)$ is constant on $X_k$, where $\kappa(p)$ denotes the curvature of $X_k$ at $p \in X_k$.*
3. *The function $g$ is given by*

$$g(x, y) = (ax^2 + 2hxy + by^2)^{d/2},$$

*where $a, b$ and $h$ satisfy $ab - h^2 \neq 0$. Thus, each $X_k$ is either a hyperbola or an ellipse centered at the origin.*

Finally, we prove the following in Section 6.

**Proposition 2.** *There exists a function $g(x, y) = f(x) + j(y)$ which satisfies the following.*

1. *Every level curve of $g$ is strictly convex with $S_g = \mathbb{R}$.*
2. *For $k \in S_g$, $\kappa(p)|\nabla g(p)|^3 = c(k)$ is constant on $X_k$, where $\kappa(p)$ denotes the curvature of $X_k$ at $p \in X_k$.*
3. *The function $g$ does not satisfy $(\mathcal{A}^*)$.*

A lot of properties of conic sections (especially, parabolas) have been proved to be characteristic ones [6–13]. For hyperbolas and ellipses centered at the origin, using the support function $h$ and the curvature function $\kappa$ of a plane curve, a characterization theorem was established [14], from which we get the proof of Theorem 3 in Section 5.

Some characterization theorems for hyperplanes, circular hypercylinders, hyperspheres, elliptic paraboloids and elliptic hyperboloids in the Euclidean space $\mathbb{E}^{n+1}$ were established in [5,15–19]. For a characterization of hyperbolic space in the Minkowski space $\mathbb{E}_1^{n+1}$, we refer to [20].

In this article, all functions are smooth ($C^{(3)}$).

## 2. Preliminaries

Suppose that $X$ is a smooth strictly convex curve in the plane $\mathbb{E}^2$ with the unit normal $N$ pointing to the convex side. For a fixed point $p \in X$ and for a sufficiently small $h > 0$, we take the line $\ell$ passing

through the point $p + hN(p)$ which is parallel to the tangent $t$ to $X$ at $p$. We denote by $A$ and $B$ the points where the line $\ell$ meets the curve $X$ and put $\mathcal{L}_p(h)$ and $\mathcal{A}_p(h)$ the length of the chord $AB$ of $X$ and the area of the region bounded by the curve and the line $\ell$, respectively.

Without loss of generality, we may take a coordinate system $(x, y)$ of $\mathbb{E}^2$ with the origin $p$, the tangent line to $X$ at $p$ is the $x$-axis. Hence $X$ is locally the graph of a strictly convex function $f : \mathbb{R} \to \mathbb{R}$ with $f(p) = 0$.

For a sufficiently small $h > 0$, we get

$$\mathcal{A}_p(h) = \int_{I_p(h)} \{h - f(x)\} dx,$$

$$\mathcal{L}_p(h) = \int_{I_p(h)} 1 dx,$$

where we put $I_p(h) = \{x \in \mathbb{R} | f(x) < h\}$ and $\mathcal{L}_p(h)$ is nothing but the length of $I_p(h)$. Note that we also have

$$\mathcal{A}_p(h) = \int_{y=0}^{h} \mathcal{L}_p(y) dy = \int_{y=0}^{h} \{\int_{I_p(y)} 1 dx\} dy,$$

from which we obtain

$$\mathcal{A}'_p(h) = \mathcal{L}_p(h).$$

We have the following [3]:

**Lemma 1.** *Suppose that $X$ is a smooth strictly convex curve in the plane $\mathbb{E}^2$. Then for a point $p \in X$ we have*

$$\lim_{t \to 0} \frac{1}{\sqrt{t}} \mathcal{L}_p(t) = \frac{2\sqrt{2}}{\sqrt{\kappa(p)}}$$

*and*

$$\lim_{t \to 0} \frac{1}{t\sqrt{t}} \mathcal{A}_p(t) = \frac{4\sqrt{2}}{3\sqrt{\kappa(p)}},$$

*where $\kappa(p)$ is the curvature of $X$ at $p$.*

Now, we consider the family of strictly convex level curves $X_k = g^{-1}(k)$ of a function $g : \mathbb{R}^2 \to \mathbb{R}$ with $k \in S_g$.

Suppose that the function $g$ satisfies condition $(\mathcal{A}^*)$. For each $k \in S_g$ and $p \in X_k$ we denote by $\kappa(p)$ the curvature of $X_k$ at $p$

By considering $-g$ if necessary, we may assume that $I_k$ is of the form $(k, a)$ with $k < a$, and hence we have $N = \nabla g / |\nabla g|$ on $X_k$. For a fixed point $p \in X_k$ and a small $t > 0$, we have

$$\mathcal{A}_p(t) = \mathcal{A}_p^*(k, h(t)) = \phi_k(h(t)),$$

where $h = h(t)$ is a function with $h(0) = 0$. Differentiating with respect to $t$ gives

$$\mathcal{L}_p(t) = \mathcal{A}'_p(t) = \phi'_k(h)h'(t),$$

where $\phi'_k(h)$ is the derivative of $\phi_k$ with respect to $h$. This shows that

$$\frac{1}{\sqrt{t}} \mathcal{L}_p(t) = \frac{\phi'_k(h)}{\sqrt{h}} \sqrt{\frac{h(t)}{t}} h'(t). \tag{1}$$

Next, we use the following lemma for the limit of $h'(t)$ as $t \to 0$.

**Lemma 2.** *We have*

$$\lim_{t \to 0} h'(t) = |\nabla g(p)|. \tag{2}$$

**Proof.** See the proof of Lemma 8 in [5]. □

It follows from (2) that

$$\lim_{t \to 0} \sqrt{\frac{h(t)}{t}} = \sqrt{|\nabla g(p)|}. \tag{3}$$

Together with Lemma 1, (2) and (3), (1) implies that $\lim_{h \to 0} \phi'_k(h)/\sqrt{h}$ exists (say, $\gamma(k)$), which is independent of $p \in X_k$. Furthermore, we also obtain

$$\kappa(p)|\nabla g(p)|^3 = \frac{8}{\gamma(k)^2},$$

which is constant on the level curve $X_k$.

Finally, we obtain the following lemma which is useful in the proof of Theorems stated in Section 1.

**Lemma 3.** *We suppose that a function* $g : \mathbb{R}^2 \to \mathbb{R}$ *satisfies condition* $(\mathcal{A}^*)$. *Then, for each* $k \in S_g$, *on* $X_k$ *the function defined by*

$$\kappa(p)|\nabla g(p)|^3 = c(k)$$

*is constant on* $X_k$, *where* $\kappa(p)$ *is the curvature of* $X_k$ *at* $p$.

**Remark 1.** *Lemma 3 is a special case* ($n = 1$) *of Lemma 8 in [5]. For conveniences, we gave a brief proof.*

## 3. Proof of Theorem 1

In this section, we give a proof of Theorem 1 stated in Section 1.

For a nonzero real number $a (\neq 1)$ and a smooth function $f : \mathbb{R} \to \mathbb{R}$, we investigate the level curves of the function $g = g_a : \mathbb{R}^2 \to \mathbb{R}$ defined by $g_a(x, y) = y^a - f(x)$.

Suppose that the function $g$ satisfies condition $(\mathcal{A}^*)$. Then, it follows from Lemma 3 that on the level curve $X_k = g^{-1}(k)$ with $k \in S_g$ we have

$$\kappa(p)|\nabla g(p)|^3 = c(k), \tag{4}$$

where $c(k)$ is a function of $k \in S_g$.

Note that for $p = (x, y) \in X_k$ with $y^a = f(x) + k$ we have

$$|\nabla g(p)|^3 = \{f'(x)^2 + a^2(f(x) + k)^{\frac{2a-2}{a}}\}^{\frac{3}{2}},$$

and hence

$$\kappa(p)|\nabla g(p)|^3 = |a^2(f(x) + k)^{\frac{2a-2}{a}} f''(x) + a(1 - a)(f(x) + k)^{\frac{a-2}{a}} f'(x)^2|. \tag{5}$$

Thus, it follows from (4) and (5) that for some nonzero $c = c(k)$ with $k \in S_g$, the function $f(x)$ satisfies

$$a^2(f(x) + k)^{\frac{2a-2}{a}} f''(x) + a(1 - a)(f(x) + k)^{\frac{a-2}{a}} f'(x)^2 = c(k),$$

which can be rewritten as

$$f''(x) + \frac{1 - a}{a}(f(x) + k)^{-1} f'(x)^2 = \frac{c(k)}{a^2}(f(x) + k)^{\frac{2-2a}{a}}. \tag{6}$$

By differentiating (6) with respect to $k$, we get

$$f'(x)^2 = \frac{c'(k)}{a(a-1)}(f(x) + k)^{\frac{2}{a}} - 2\frac{c(k)}{a^2}(f(x) + k)^{\frac{2-a}{a}}. \tag{7}$$

Putting $u = f(x) + k$ and $v = du/dx = f'(x)$, we get from (6)

$$\frac{dv}{du} + \frac{1-a}{a} u^{-1} v = \frac{c(k)}{a^2} u^{\frac{2-2a}{a}} v^{-1},$$

(8)

which is a Bernoulli equation. By letting $w = v^2$, we obtain

$$\frac{dw}{du} + \frac{2-2a}{a} u^{-1} w = \frac{2c(k)}{a^2} u^{\frac{2-2a}{a}}.$$

(9)

Since $u^{\frac{2-2a}{a}}$ is an integrating factor of (9), we get

$$\frac{d}{du}(wu^{\frac{2-2a}{a}}) = \frac{2c(k)}{a^2} u^{\frac{4-4a}{a}}.$$

(10)

Now, in order to integrate (10), we divide by some cases as follows.

**Case 1.** Suppose that $a = \frac{4}{3}$. Then, from (10) we have

$$w = \{\frac{9c(k)}{8} \ln u + b(k)\} \sqrt{u},$$

(11)

where $b = b(k)$ is a constant. Since $u = f(x) + k$ and $w = f'(x)^2$, (7) and (11) show that

$$c(k) \ln(f(x) + k) + \frac{8}{9} b(k) = 2c'(k)(f(x) + k) - c(k).$$

(12)

By differentiating (12) with respect to $x$, we obtain

$$2c'(k)(f(x) + k) = c(k).$$

(13)

Since $c(k)$ is nonzero, (13) leads to a contradiction.

**Case 2.** Suppose that $a \neq \frac{4}{3}$. Then, from (8) we have

$$w = a(k)u^\alpha + b(k)u^\beta, a(k) = \frac{2c(k)}{(4-3a)a}, \alpha = \frac{2-a}{a}, \beta = \frac{2a-2}{a},$$

(14)

where $b = b(k)$ is a constant. Since $u = f(x) + k$ and $w = f'(x)^2$, it follows from (7) and (14) that

$$b(k)(f(x) + k)^{\frac{3a-4}{a}} = \frac{c'(k)}{a(a-1)}(f(x) + k) - 4c(k)\frac{a-2}{a^2(3a-4)}.$$

(15)

By differentiating (15) with respect to $x$, we get

$$b(k)(f(x) + k)^{\frac{2a-4}{a}} = \frac{c'(k)}{a(a-1)}.$$

(16)

If $b(k) \neq 0$, then (16) shows that $a = 2$. If $b(k) = 0$, then it follows from (15) and (16) that $c'(k) = 0$, and hence $a = 2$.

Finally, we consider the remaining case as follows.

**Case 3.** Suppose that $a = 2$. Then, it follows from (7) that for the constant $c = c(k)$

$$f'(x)^2 = \frac{c'(k)}{2}(f(x) + k) - \frac{c(k)}{2}.$$

(17)

If $c'(k) = 0$, that is, $c$ is independent of $k$, then (17) shows that $f(x)$ is a linear function. Hence each level curve $X_k$ of the function $g(x, y) = y^2 - f(x)$ is a parabola. If $c'(k) \neq 0$, then differentiating both sides of (17) with respect to $x$ shows

$$4f''(x) = c'(k).$$

This yields that $f(x)$ is a quadratic function and $c(k)$ is a linear function in $k$.

Combining Cases 1–3, we proved the following:

$$1) \Rightarrow 2) \Rightarrow 3).$$

Conversely, suppose that the function $g$ is given by

$$g(x, y) = y^2 - (ax^2 + bx + c),$$

where $a, b$ and $c$ are constants with $a^2 + b^2 \neq 0$. Then, each level curve $X_k$ of $g$ is an ellipse ($a < 0$), a hyperbola ($a > 0$) or a parabola ($a = 0, b \neq 0$). It follows from Section 1 or [4], pp. 6–7 that the function $g$ satisfies condition $(\mathcal{A}^*)$.

This shows that Theorem 1 holds.

**Remark 2.** *It follows from the proof of Theorem 1 that the constant $c = c(k)$ is independent of $k$ if $g(x, y) = y^2 - 4ax, a \neq 0$ and it is a linear function in $k$ if $g(x, y) = y^2 - ax^2, a \neq 0$.*

Finally, we note the following.

**Remark 3.** *Suppose that a smooth function $g : \mathbb{R}^2 \to \mathbb{R}$ satisfies condition $(\mathcal{A}^*)$ with*

$$\kappa(p)|\nabla g(p)|^3 = c(k),$$

*where $p \in X_k = g^{-1}(k)$ and $k \in S_g$. Then for any positive constant $d$, there exists a composite function $G = \phi \circ g$ satisfying condition $(\mathcal{A}^*)$ with*

$$\kappa(p)|\nabla G(p)|^3 = d. \tag{18}$$

*Note that the function $G = \phi \circ g$ has the same level curves as the function $g$.*

*In order to prove (18), we denote by $\phi(t)$ an indefinite integral of the function $(d/c(t))^{1/3}$. Then for $p \in G^{-1}(k) = g^{-1}(\phi^{-1}(k))$ we get*

$$|\nabla G(p)| = \phi'(g(p))|\nabla g(p)|.$$

*Hence, on each level curve $G^{-1}(k) = g^{-1}(\phi^{-1}(k))$ we obtain*

$$\kappa(p)|\nabla G(p)|^3 = c(\phi^{-1}(k))\phi'(\phi^{-1}(k))^3 = d.$$

## 4. Proof of Theorem 2

In this section, we give a proof of Theorem 2.

We consider a function $g$ defined by $g(x, y) = f(x) + j(y)$ for some functions $f(x)$ and $j(y)$. Then at the point $p \in X_k = g^{-1}(k)$ we have

$$|\nabla g(p)|^3 = \{f'(x)^2 + j'(y)^2\}^{\frac{3}{2}},$$
$$\kappa(p)|\nabla g(p)|^3 = |f''(x)j'(y)^2 + f'(x)^2 j''(y)|.$$

Suppose that the function $g$ satisfies condition $(\mathcal{A}^*)$. Then, it follows from Lemma 3 that on the level curve $X_k : f(x) + j(y) = k$ we get for some nonzero constant $c = c(k)$

$$f''(x)j'(y)^2 + f'(x)^2 j''(y) = c(k), \tag{19}$$

which shows that the set $V = \{(x,y) \in X_k | f'(x) = 0 \quad \text{or} \quad j'(y) = 0\}$ has no interior points in the level curve $X_k$. Hence by continuity, without loss of generality we may assume that $V$ is empty.

First, we consider $y$ as a function of $x$ and $k$. Then, we rewrite (19) as follows

$$f''(x) + f'(x)^2 \frac{j''(y)}{j'(y)^2} = \frac{c(k)}{j'(y)^2}, \quad j(y) + f(x) = k. \tag{20}$$

Putting $u = -f(x) + k$ and $v = du/dx = -f'(x)$, we get

$$\frac{dv}{du} - \frac{j''(y)}{j'(y)^2} v = -\frac{c(k)}{j'(y)^2} v^{-1},$$

which is a Bernoulli equation. By letting $w = v^2 = f'(x)^2$, we obtain

$$\frac{dw}{du} - \frac{2j''(y)}{j'(y)^2} w = -\frac{2c(k)}{j'(y)^2}. \tag{21}$$

Since $u = j(y)$, we see that $j'(y)^{-2}$ is an integrating factor of (21). Hence we get

$$\frac{d}{du}(wj'(y)^{-2}) = -2c(k)j'(y)^{-4}.$$

Thus we obtain

$$f'(x)^2 = w = -2c(k)j'(y)^2\{\phi(y) + d(k)\}, \tag{22}$$

where $\phi(y)$ is a function of $y$ satisfying $\phi'(y) = j'(y)^{-3}$ and $d = d(k)$ is a constant.

On the other hand, by differentiating (20) with respect to $k$, we get

$$f'(x)^2\{j'(y)j'''(y) - 2j''(y)^2\} = c'(k)j'(y)^2 - 2c(k)j''(y). \tag{23}$$

It follows from (22) and (23) that

$$a(k)j'(y)^2 - j''(y) = j'(y)^2\{2j''(y)^2 - j'(y)j'''(y)\}\{\phi(y) + d(k)\}, \tag{24}$$

where we use $a(k) = \frac{c'(k)}{2c(k)}$. Or equivalently, we get

$$\phi(y) + d(k) = \frac{a(k)j'(y)^2 - j''(y)}{j'(y)^2\{2j''(y)^2 - j'(y)j'''(y)\}}, \tag{25}$$

where the denominator does not vanish. Even though $j(y)$ was assumed to be $C^{(3)}$, (24) implies that the function $\{2j''(y)^2 - j'(y)j'''(y)\}$ is differentiable. By differentiating (25) with respect to $x$, it is straightforward to show that

$$\{a(k)j'(y)^2 - j''(y)\}\frac{d}{dy}\{2j''(y)^2 - j'(y)j'''(y)\} = 0. \tag{26}$$

Together with (24), (26) yields that $2j''(y)^2 - j'(y)j'''(y)$ is constant. Hence, for some constant $\alpha$ we have

$$2j''(y)^2 - j'(y)j'''(y) = \alpha. \tag{27}$$

Next, interchanging the role of $x$ and $y$ in the above discussions, we consider $x$ as a function of $y$ and $k$. Then, (22) gives

$$j'(y)^2 = -2c(k)f'(x)^2\{\psi(x) + e(k)\}, \tag{28}$$

where $\psi(x)$ is a function of $x$ satisfying $\psi'(x) = f'(x)^{-3}$ and $e = e(k)$ is a constant. In the same argument as the above, we obtain the corresponding equations from (23)–(27). For example, we get from (26)

$$\{a(k)f'(x)^2 - f''(x)\}\frac{d}{dx}\{2f''(x)^2 - f'(x)f'''(x)\} = 0. \tag{29}$$

Thus, for some constant $\beta$, we also get

$$2f''(x)^2 - f'(x)f'''(x) = \beta. \tag{30}$$

By integrating (24) and (30) respectively, we obtain for some constants $\gamma$ and $\delta$

$$2j''(y)^2 = \gamma j'(y)^4 + \alpha \tag{31}$$

and its corresponding equation

$$2f''(x)^2 = \delta f'(x)^4 + \beta. \tag{32}$$

Differentiating (19) with respect to $x$, we have

$$\frac{1}{j'(y)}\{f'''(x)j'(y)^3 - f'(x)^3 j'''(y)\} = \frac{d}{dx}\{f''(x)j'(y)^2 + f'(x)^2 j''(y)\} = 0. \tag{33}$$

Together with (31) and (32), this shows that $j(y)$ is quadratic in $y$ if and only if $f(x)$ is quadratic in $x$.

Hereafter, we assume that neither $f(x)$ nor $j(y)$ are quadratic. Then, combining (27), (30), (31) and (32), it follows from (33) that

$$(\gamma - \delta)f'(x)^4 j'(y)^4 = 0,$$

which shows that $\gamma = \delta$. Hence, for a nonzero constant $\gamma$ the functions $f(x)$ and $j(y)$ satisfy, respectively

$$2f''(x)^2 = \gamma f'(x)^4 + \beta \tag{34}$$

and

$$2j''(y)^2 = \gamma j'(y)^4 + \alpha. \tag{35}$$

Differentiating (34) and (35) with respect to $x$ and $y$, respectively, implies

$$f'''(x) = \gamma f'(x)^3, \quad j'''(y) = \gamma j'(y)^3, \tag{36}$$

where $\gamma$ is a nonzero constant.

Conversely, we prove the following for later use in Section 6.

**Lemma 4.** *Suppose that the functions $f(x)$ and $j(y)$ satisfy (34) and (35) for some constants $\alpha$ and $\beta$, respectively. Then on each level curve $X_k$ with $k \in S_g$ of the function $g(x, y) = f(x) + j(y)$, $\kappa(p)|\nabla g(p)|^3$ is constant.*

**Proof.** Using (36), it follows from the first equality of (33) that on the level curve $X_k$ of the function $g$, we have

$$\frac{d}{dx}\{f''(x)j'(y)^2 + f'(x)^2 j''(y)\} = 0.$$

This completes the proof of Lemma 4. $\square$

Finally, we proceed on our way. We divide by two cases as follows.

**Case 1.** Suppose that $j(y)$ is a polynomial of degree $\deg h = n \geq 3$. Then, by counting the degree of both sides of the second equation in (36) we see that the constant $\gamma$ must vanish. This contradiction shows that the polynomial $j(y)$ is quadratic.

**Case 2.** Suppose that $j(y)$ is a rational function given by

$$j(y) = \frac{s(y)}{q(y)},$$

where $q$ and $s$ are relatively prime polynomials of degree $\deg q = m(\geq 1)$ and $\deg s = n(\geq 0)$, respectively.

**Subcase 2-1.** Suppose that $m \geq n$. Then we get from (35) that

$$\alpha q(y)^8 = \gamma A(y)^4 - 2B(y)^2, \tag{37}$$

where we put

$$A(y) = s'(y)q(y) - s(y)q'(y), B(y) = A'(y)q(y)^2 - 2q(y)q'(y)A(y).$$

Since the degree of the right hand side of (37) is less than or equal to $8m - 4$, (37) shows that $\alpha$ must vanish. By integrating (30′) with $\alpha = 0$, we obtain for some constant $a$ and $b$

$$j(y) = \frac{1}{a} \ln |ay + b|,$$

which is a contradiction.

**Subcase 2-2.** Suppose that $m \leq n - 2$. We put

$$j(y) = \frac{s(y)}{q(y)} = r(y) + \frac{t(y)}{q(y)},$$

where $\deg r = a = n - m \geq 2$ and $\deg t \leq m - 1$. Then we get from (30′) that

$$\gamma \{r'(y)q(y)^2 + A(y)\}^4 = 2\{r''(y)q(y)^4 + B(y)\}^2 - \alpha q(y)^8, \tag{38}$$

where we put

$$A(y) = t'(y)q(y) - t(y)q'(y), B(y) = A'(y)q(y)^2 - 2q(y)q'(y)A(y).$$

Since the degree of the left hand side of (38) is $8m + 4a - 4$ and the degree of the right hand side of (38) is less than or equal to $8m + 2a - 4$, we see that $\gamma$ must vanish, which is a contradiction. Hence this case cannot occur.

**Subcase 2-3.** Suppose that $m = n - 1$. Then we have $r(y) = r_0 y + r_1$ with $r_0 \neq 0$,

$$j'(y) = r_0 + \frac{A(y)}{q(y)^2}, \quad A(y) = t'(y)q(y) - t(y)q'(y),$$

$$j''(y) = \frac{\bar{B}(y)}{q(y)^3}, \quad \bar{B}(y) = A'(y)q(y) - 2A(y)q'(y)$$

and

$$j'''(y) = \frac{C(y)}{q(y)^6}, \quad C(y) = \bar{B}'(y)q(y)^3 - 3\bar{B}(y)q(y)^2 q'(y).$$

It follows from the second equation of (36) that

$$\gamma\{r_0 q(y)^2 + A(y)\}^3 = C(y).$$

Note that the left hand side is of degree $6m$, but the right hand side is of degree $\deg C \le 6m - 4$. Hence, the constant $\gamma$ must vanish, which is a contradiction. Thus, this case cannot occur.

Combining Cases 1 and 2, we see that the function $j(y)$ is a quadratic polynomial. Therefore, Theorem 1 completes the proof of Theorem 2.

## 5. Proof of Theorem 3

In this section, we give a proof of Theorem 3.

Consider a smooth homogeneous function $g : \mathbb{R}^2 \to \mathbb{R}$ of degree $d$. Suppose that the function $g$ satisfies $(\mathcal{A}^*)$. Then, it follows from Lemma 3 that on the level curve $X_k = g^{-1}(k)$ with $k \in S_g$ we have

$$\kappa(p)|\nabla g(p)|^3 = c(k), \tag{39}$$

where $c(k)$ is a nonzero function of $k \in S_g$.

We recall the support function $h(p)$ on the level curve $X_k$, which is defined by

$$h(p) = \langle p, N(p) \rangle,$$

where $N(p)$ denotes the unit normal to $X_k$. Note that the unit normal $N(p)$ to $X_k$ is given by

$$N(p) = \frac{\nabla g(p)}{|\nabla g(p)|}.$$

Since the function $g$ is homogeneous of degree $d$, by the Euler identity, on $X_k$ we obtain

$$h(p) = \frac{\langle p, \nabla g(p) \rangle}{|\nabla g(p)|} = \frac{dk}{|\nabla g(p)|}. \tag{40}$$

Thus, it follows from (39) and (40) that $X_k$ satisfies

$$\kappa(p) = \frac{c(k)}{(dk)^3} h(p)^3.$$

Now, we use the following characterization theorem [14].

**Proposition 3.** *Suppose that $X$ is a smooth curve in the plane $\mathbb{E}^2$ of which curvature $\kappa$ does not vanish identically. Then $X$ satisfies for some constant $c$*

$$\kappa(p) = ch(p)^3.$$

*if and only if $X$ is a connected open arc of either a hyperbola or an ellipse centered at the origin.*

The above proposition shows that for each $k \in S_g$, the level curve $X_k$ is either a hyperbola centered at the origin or an ellipse centered at the origin. Without loss of generality, we may assume that $1 \in S_g$. Then, the level curve $X_1 = g^{-1}(1)$ is given by

$$ax^2 + 2hxy + by^2 = 1, \tag{41}$$

where $a, b$ and $h$ satisfy $ab - h^2 \ne 0$.

We claim that

$$g(x, y) = (ax^2 + 2hxy + by^2)^{d/2}. \tag{42}$$

where $a, b$ and $h$ satisfy $ab - h^2 \neq 0$.

In order to prove (42), for a fixed point $p = (x, y) \in \mathbb{R}^2$ we let $g(x, y) = k$, that is, $p = (x, y) \in X_k$. Then we have for $t = k^{-1/d}$

$$g(tx, ty) = 1.$$

Hence we get from (41)

$$ax^2 + 2hxy + by^2 = t^{-2} = k^{2/d}.$$

This shows that

$$g(x, y) = k = (ax^2 + 2hxy + by^2)^{d/2},$$

which proves the above mentioned claim. Therefore, the proof of Theorem 3 was completed.

## 6. Proof of Proposition 2

In this section, we prove Proposition 2.

We denote by $\psi(t)$ the function defined by

$$\psi'(t) = \frac{1}{\sqrt{1 + t^4}}, \quad \psi(0) = 0$$

and we put

$$a = \int_0^\infty (t^4 + 1)^{-1/2} dt.$$

Then, both of $\psi : (-\infty, \infty) \to (-a, a)$ and $\psi^{-1} : (-a, a) \to (-\infty, \infty)$ are strictly increasing odd functions.

Now, we consider the function $g(x, y) = f(x) + j(y)$ defined on the domain $U = (0, a) \times (0, \infty) \subset \mathbb{R}^2$ with $j(y) = \ln y$ and

$$f(x) = \ln \psi^{-1}(x).$$

Then we have $S_g = R_g = \mathbb{R}$ and $I_k = (k, \infty)$. Furthermore, it is straightforward to show that the functions $f(x)$ and $j(y)$ satisfies (34) and (30′) respectively, where we put $\gamma = 2, \alpha = 0$ and $\beta = -8$. Thus, Lemma 4 implies that on each level curve $X_k$ of the function $g(x, y) = f(x) + j(y)$, $\kappa(p)|\nabla g(p)|^3$ is constant.

However, we show that the function $g$ cannot satisfy condition $(\mathcal{A}^*)$ as follows. For each $k \in S_g = \mathbb{R}$, the level curve $X_k = g^{-1}(k)$ of $g$ are given by

$$y\psi^{-1}(x) = e^k, \quad x, y > 0.$$

Note that $X_k$ is the graph of the strictly convex function given by

$$y = \frac{e^k}{\psi^{-1}(x)}, \quad x \in (0, a),$$

which satisfies

$$\frac{dy}{dx} < 0, \quad \frac{d^2y}{dx^2} > 0$$

and

$$\lim_{x \to 0} y = \infty, \quad \lim_{x \to 0} \frac{dy}{dx} = -\infty, \quad \lim_{x \to a} y = 0, \quad \lim_{x \to a} \frac{dy}{dx} = -e^k.$$

Hence, each level curve $X_k$ approaches the point $(a, 0)$ and the $y$-axis is an asymptote of $X_k$. For a fixed point $v$ of $X_0$ and a negative number $h < 0$, let $p \in X_h$ be the point where the tangent $t$ to $X_h$ is parallel to the tangent $\ell$ to $X_0$ at $v$. We denote by $A(h)$ and $B(h)$ the points where the tangent $\ell$ to $X_0$ at $v$ intersects the level curve $X_h$.

Suppose that the function $g$ satisfies condition $(\mathcal{A}^*)$. Then, the area of the region enclosed by $X_h$ and the chord $A(h)B(h)$ of $X_h$ is $\mathcal{A}_p^*(h, -h) = \phi_h(-h)$, which is independent of $v$. We also denote by $A$ and $B$ the points where the tangent $\ell$ to $X_0$ at $v$ meets the coordinate axes, respectively. Then, $A(h)$ and $B(h)$ tend to $A$ and $B$, respectively, as $h$ tends to $-\infty$. Furthermore, as $h$ tends to $-\infty$, $\phi_h(-h)$ goes to the area of the triangle $OAB$, where $O$ denotes the origin. Thus, the area of the triangle $OAB$ is independent of the point $v \in X_0$. This contradicts the following lemma, which might be well known. Therefore the function $g(x, y) = f(x) + j(y)$ does not satisfy condition $(\mathcal{A}^*)$. This gives a proof of Proposition 2.

**Lemma 5.** *Suppose that $X$ denotes the graph of a strictly convex function $f : I \to \mathbb{R}$ defined on an open interval $I$. Then $X$ satisfies the following condition $(A)$ if and only if $X$ is a part of the hyperbola given by $xy = c$ for some nonzero $c$.*

$(A)$*: For a point $v \in X$, we put $A$ and $B$ at the points where the tangent $\ell$ to $X$ at $v$ intersects coordinate axes, respectively. Then the area of the triangle $OAB$ is independent of the point $v \in X$.*

**Proof.** Suppose that $X$ satisfies condition $(A)$. Then, $f'(x)$ vanishes nowhere on the interval $I$. For a point $v = (x, f(x))$, the area $A(x)$ of the triangle $OAB$ is given by

$$A(x) = \frac{-1}{2f'(x)} \{xf'(x) - f(x)\}^2. \tag{43}$$

Differentiating (43) with respect to $x$ gives

$$\frac{-1}{2f'(x)^2} \{x^2 f'(x)^2 - f(x)^2\} f''(x) = 0. \tag{44}$$

By assumption, $f''(x) > 0$. Hence, we get from (44)

$$x^2 f'(x)^2 - f(x)^2 = 0,$$

which shows that $X$ is a hyperbola given by $xy = c$ for some nonzero $c$.

It is trivial to prove the converse. $\square$

**Remark 4.** *For some higher dimensional analogues of Lemma 5, see [19].*

**Author Contributions:** D.-S.K. and Y.H.K. set up the problem and computed the details and Y.-T.J. checked and polished the draft.

**Funding:** The first named author was supported by the Basic Science Research Program through the National Research Foundation of Korea (NRF) funded by the Ministry of Education (NRF-2018R1D1A3B05050223). The second named author was supported by the National Research Foundation of Korea (NRF) Grant funded by the Korea Government (MSIP) grant number 2016R1A2B1006974.

**Acknowledgments:** We would like to thank the referee for the careful review and the valuable comments to improve the paper.

**Conflicts of Interest:** The authors declare no conflict of interest.

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
