# Peer review of "Area Properties of Strictly Convex Curves"

_mathematics, doi:10.3390/math7050391_

Round 1

Reviewer 1 Report

Referee report on the manuscript

Area properties of strictly convex curve

By Dong-Soo Kim, Young Ho Kim and Yoon-Tae Jung

There are interesting geometric results in this paper. The material is presented in a good expository style. The subject of the paper is clearly defined.

The presented work meet the requirements for a well-motivated and well-written paper.

Therefore, I recommend this manuscript for publication in “Mathematics”.

Author Response

Thank you very for for reviewing my paper which is worth being  published.

Many thanks for your work.

Sincerely yours

Young Ho Kim

Reviewer 2 Report

I propose Authors to add to the abstract some sentences about what kind of characterization are used (for example equiaffine etc)

I think that Authors should add some graphs (using Mathematica or Matlap)

Author Response

The reviewer asked us to add in the abstract some sentences about what kind of characterization are used and proposed to add some graphs.

Regarding his /her comments, we add some back ground of the problem we are dealing with and a sentence to mention some characterization theorems of conic sections in the abstract.

Also, in Page 2, we add a picture for the problem on equiaffine transformation.

This manuscript is a resubmission of an earlier submission. The following is a list of the peer review reports and author responses from that submission.

Round 1

Reviewer 1 Report

I have some critical remarks:

1) There are numbered equalities that are not used in the text.

2) In Theorem 1.4. the notation h is used as the variable and in the same time as the function. 

3) I think that the parameter p is used instead of a in a line 212.

Reviewer 2 Report

The paper under review provides some good characterizations for conic sections in the Euclidean plane. The results are correct and the paper is well written. However, I feel that the results may not meet the standard of the journal Mathematics. So I have to suggest rejecting it but encourage the authors to find a more suitable journal to publish it.